# A diagnostic real-time PCR assay for the rapid identification of the tomato-potato psyllid, *Bactericera cockerelli* (Šulc, 1909) and development of a psyllid barcoding database

J. C. Sumner-Kalkun[1]*, M. J. Sjölund[1], Y. M. Arnsdorf[1], M. Carnegie[1], F. Highet[1], D. Ouvrard[2,3], A. F. C. Greenslade[4], J. R. Bell[4], R. Sigvald[5], D. M. Kenyon[1]

**1** SASA, Edinburgh, United Kingdom, **2** Department of Life Sciences, Natural History Museum, London, United Kingdom, **3** Entomology and invasive plants Unit, Plant Health Laboratory, ANSES, Montferrier-sur-Lez Cedex, France, **4** Rothamsted Insect Survey, Rothamsted Research, Harpenden, Hertfordshire, United Kingdom, **5** Department of Ecology, Swedish University of Agricultural Sciences, Uppsala, Sweden

* jason.sumner-kalkun@sasa.gov.scot

## Abstract

The accurate and rapid identification of insect pests is an important step in the prevention and control of outbreaks in areas that are otherwise pest free. The potato-tomato psyllid *Bactericera cockerelli* (Šulc, 1909) is the main vector of 'Candidatus Liberibacter solanacearum' on potato and tomato crops in North America and New Zealand; and is considered a threat for introduction in Europe and other pest-free regions. This study describes the design and validation of the first species-specific TaqMan probe-based real-time PCR assay, targeting the ITS2 gene region of *B. cockerelli*. The assay detected *B. cockerelli* genomic DNA from adults, immatures, and eggs, with 100% accuracy. This assay also detected DNA from cloned plasmids containing the ITS2 region of *B. cockerelli* with 100% accuracy. The assay showed 0% false positives when tested on genomic and cloned DNA from 73 other psyllid species collected from across Europe, New Zealand, Mexico and the USA. This included 8 other species in the *Bactericera* genus and the main vectors of 'Candidatus Liberibacter solanacearum' worldwide. The limit of detection for this assay at optimum conditions was 0.000001ng DNA (~200 copies) of ITS2 DNA which equates to around a 1:10000 dilution of DNA from one single adult specimen. This assay is the first real-time PCR based method for accurate, robust, sensitive and specific identification of *B. cockerelli* from all life stages. It can be used as a surveillance and monitoring tool to further study this important crop pest and to aid the prevention of outbreaks, or to prevent their spread after establishment in new areas.

## 1. Introduction

The psyllid *Bactericera cockerelli* (Šulc, 1909), (commonly known as "Potato Psyllids" or "Tomato-Potato Psyllid"), is a major pest of cultivated Solanaceous crops including potato and tomato [1]. Feeding by this psyllid causes severe damage to potato plants including: deformed tubers; production of numerous small, poor quality tubers; curling of leaves and petioles; and

**Data Availability Statement:** All Co1 and ITS2 sequences area available from GenBank accession numbers (MT021761-MT021824; MT027551-MT027599; MT038907-MT038996; MT040955-MT040966). These will be made accessible on request.

**Funding:** DK: This work was supported by the EU Horizon2020 Programme under grant agreement No. 635646, POnTE (Pest Organisms Threatening Europe) https://ec.europa.eu/programmes/horizon2020/en and the Scottish Government [RRL/001/14]. JB: The Rothamsted Insect Survey, a National Capability, is funded by the Biotechnology and Biological Sciences Research Council under the Core Capability Grant BBS/E/C/000J0200. https://bbsrc.ukri.org/funding/ The funders had no role in study design, data collection and analysis, decision to publish, or preparation of the manuscript.

**Competing interests:** The authors have declared that no competing interests exist.

yellowing or purpling of leaves. This leads to stunted growth and loss of yield [2]. *Bactericera cockerelli* is also the main vector of '*Candidatus* Liberibacter solanacearum' (Lso) which is associated with Zebra Chip in Central and North America and New Zealand [3–8]. *Bactericera cockerelli* is thought to originate from South-Western USA and Mexico [2,9] and from here has spread via natural and human-mediated dispersal to extend its range [10]. Outside America it is now established in New Zealand [11] and more recently Western Australia [12].

While *B. cockerelli* prefers to complete its life cycle on Solanaceous plants it can also complete development on species of Convolvulaceae (Bindweeds and Morning Glories) [13]. In addition, adult *B. cockerelli* have been found on over 40 species belonging to 20 families, however most of these are either casual, food or shelter plants on which the psyllid is unable to complete a full life cycle [2,9,14–19]. Four biotypes of *B. cockerelli* have been described according to polymorphisms in the mitochondrial cytochrome *c oxidase subunit I* (COI) gene and represent geographically distinct populations; central, western, north-western, and south-western [20,21]. Evidence suggest that these genetic types may differ in their ability to spread Lso [21,22].

The phloem-limited bacterium '*Candidatus* Liberibacter solanacearum' (Lso) is a pathogen associated with Zebra Chip disease of potatoes [3,23–25] and disease in other Solanaceous crops such as cultivated tomato [1,3,26,27], pepper [28], eggplant [29], tobacco [30,31] and tomatillo [26]. Currently, *B. cockerelli* is the main vector of Lso in field and glasshouse-grown Solanaceous plants in the United States, Mexico, areas of Central America [27–30], Canada [32], New Zealand [5,6,25] and recently Ecuador [33]. Ten Lso haplotypes have been described, only three of which are associated with disease in Solanaceous plants. Haplotypes A, B, and F are associated with Zebra chip disease in America [3,34,35], whereas only haplotype A has been found in New Zealand [5,36]. Haplotype B has also been found in *Bactericera maculipennis* (Crawford) [37]. The remaining haplotypes are not vectored by *B. cockerelli* but by closely related species in the Triozidae family.

The impact of *B. cockerelli* and associated Lso transmission on agriculture is significant. Since its arrival in New Zealand circa 2005 via human-mediated dispersal it has caused millions of dollars of economic losses [6,21]. Similarly, management of *B. cockerelli* in the US is reported to have cost millions of dollars per year in major potato growing areas such as Texas [38] and the Pacific Northwest [39]. The introduction of *B. cockerelli* into potato growing regions in Europe or Asia would be devastating to the agricultural industry of those regions. If *B. cockerelli*, or a sufficient vector of Solanaceous Lso haplotypes, were to invade Europe it is estimated that the effects of Lso damage on potato and tomato would cost € 222 million per year and the negative impact of social welfare could cost an additional estimated € 114 million [40].

Currently, *B. cockerelli* is considered an A1 quarantine pest in the EPPO region [4]. Consignments of aubergine and *Capsicum* from Mexico infested with immature and adult stages of *B. cockerelli* were intercepted four times during UK border inspections between 2017–2018; indicating that there is a real threat of this pest making an incursion into the EPPO region if not properly monitored [41]. Monitoring and prevention of the spread of *B. cockerelli* is essential to prevent the risk of an outbreak of Lso on potato, tomato and other Solanaceous crops in areas where it is not currently found [42]. There is therefore an evident need for a rapid and accurate diagnostic test to identify *B. cockerelli* at all life stages not only as a tool to support import inspections, but also to assist monitoring, eradication and control strategies.

We designed a species-specific real-time PCR diagnostic assay to detect all life-stages of *B. cockerelli*, eggs, immatures and adults. The assay provides a rapid diagnostic test to quickly determine the presence of *B. cockerelli*, allowing for the early detection of invasions/introductions and aiding in the prevention of spread of this psyllid.

## 2. Materials and methods

### 2.1. Specimen collection

The assay was tested on 28 target adults *B. cockerelli* specimens and 73 non-target species consisting of 110 specimens see results section 3.1 for more info on samples. The classification follows Burckhardt & Ouvrard [43], and a complete taxonomic account of each species is given in Ouvrard [20]. Psyllid identifications were confirmed against reference type specimens in the NHM London collections. To account for intraspecific genetic variation, we obtained *B. cockerelli* specimens from Mexico (Universidad Autónoma Agraria Antonio Narro) and USA (USDA, Agricultural Research Services) from colony collections of each of the four recognised biotypes of *B. cockerelli* in Central America, the Central, Western, Northwestern, and Southwestern biotypes [19]. Specimens of *B. cockerelli* were also obtained from New Zealand lab-reared colonies (Plant Research, New Zealand). Non-target specimens were mainly obtained from 12.2 m suction-traps in the United Kingdom that form part of the Rothamsted Insect Survey network described here [44]. Specimens were also obtained from suction-traps in Finland, Germany, Spain and Sweden; as well as from field collections from Finland, Israel, Mexico, Serbia, Spain, UK and USA. Non-target specimens from different regions of the USA were used to test assay specificity on species that are commonly found in the same region as *B. cockerelli*. As immatures and eggs are the most likely life stages that inspectors might find on imported plant material, we also tested the assay on DNA extracted from immatures and eggs from Mexico and the USA for validation.

### 2.2. DNA extraction, PCR, and DNA sequencing for identification of psyllids

DNA for sequencing and assay validation was extracted from psyllids using a non-destructive method first described in [45] and adapted from [46]. Psyllid specimens were preserved in 95% Ethanol: 5% Glycerol solution. Using a 15mm long, 0.15mm diameter stainless steel entomological head-less pin (A3 size, Watkins and Doncaster) mounted in a holder, specimens were initially pierced fully through the abdomen and half-way through the thorax from the dorsal side while attempting to minimise damage to head, legs, wings, terminalia and other body parts that are used for taxonomic identification. Pierced specimens were placed in a microcentrifuge tube containing 180 μl of ATL buffer and 20 μl of proteinase-k as outlined in the DNeasy Blood and Tissue Kit from Animal Tissues (Qiagen). Samples were placed in a shaking incubator over-night (~8–10 hrs) at 56˚C at 300 rpm. The subsequent steps from the above mentioned protocol were followed and the psyllid integument voucher specimen was stored in 95% Ethanol: 5% Glycerol for morphological identification. Psyllids were DNA barcoded using one or two gene regions. The internal transcribed spacer 2 (ITS2) and cytochrome c oxidase subunit 1 (CO1) were amplified and sequenced for identification of different psyllid species. For amplification of ITS2 primers CA55p8sFcm-F and CA28sB1d-R [47] were used; and for amplification of CO1 gene regions arthropod barcoding Primers LCO1490 and HCO2198 [48] were used. All reactions were performed in 20 μl consisting of: 10 μl 2x Type-It Microsatellite PCR Kit Master Mix (Qiagen); 0.2 μM each forward and reverse primer; 7.2 μl molecular grade water (Sigma-Aldrich) and 2 μl of psyllid template DNA. Reactions were run on a Veriti 96-well thermal cycler (Applied Biosystems) using the following programs. ITS2: 95˚C for 5 mins; 25 x cycles of (95˚C for 30 s, 56˚C for 90 s, 72˚C for 30 s); and a final extension at 72˚C for 10 mins. CO1: 94˚C for 5 mins; 5 x cycles of (94˚C for 30s, 45˚C for 30s, 72˚C for 1 min); 25 x cycles of (94˚C for 30s, 51˚C for 1 min, 72˚C for 1 min); and a final extension of 72˚C for 10 mins. PCR amplified gene regions were cleaned-up using EXO-SAP and

Ethanol precipitation, then sequenced using the BigDye Terminator Cycle Sequencing Kit (Applied Biosystems), forward and reverse complimentary DNA strands were sequenced separately for each sample and analysed using a 3500xL Genetic Analyser (Applied Biosystems).

## 2.3. Bioinformatics and real-time PCR assay design

Sequence editing, assembly and alignment were performed on ".AB1" trace files uploaded to Geneious R11 v 11.1.5 (Biomatters Ltd.). Contigs were assembled after trimming sections of low-quality sequence and aligning the complimentary strands using CLUSTAL-W multiple sequence alignment method [49]. Final contigs for each species and each gene region were aligned to identify variable areas suitable as targets for *B. cockerelli* specific primer and probe sets. Primers and probes were designed using manual selection of target-specific regions analysed using the "Basic Local Alignment Search Tool" (BLAST) [50] against the NCBI GenBank database [51] and processing of selected regions for suitability/ specificity in "Primer3" [52] and "Primer-BLAST" software [53]. Primer annealing temperature, hairpin formation, self-complementarity, GC content and were assessed using "Primer3" [52]. Potential amplification of non-specific insect species was checked using Primer BLAST which includes all psyllid species present in the GenBank database. Primer and probe sets were selected/rejected based on the following parameters: primer annealing temperature 59–62˚C; primer annealing temperature + 8–10˚C for probe annealing temperature; no more than 2˚C difference in annealing temperature between primers, max probe length 30bp, no more than 3 Gs in a row in probe, amplicon length max 300bp and specificity to *B. cockerelli*.

## 2.4. Real-time PCR set-up and standards

To calculate standard curves DNA standards of *B. cockerelli* were prepared using dilution series of linearized cloned plasmid DNA. DNA was extracted as above using the non-destructive method, amplified and cloned into competent *Escherichia coli* cells using the TOPO TA cloning kit (Thermo-Fisher). DNA from successfully transformed colonies was extracted using "PureYield Plasmid Miniprep System" (Promega). For assay validation ITS2 DNA was cloned from other psyllid species (see results section 3.1). Stock DNA 10 ng/μl was linearised from cloned plasmid DNA using EcoRI restrictions enzyme (New England Biolabs), 0.5 μl of enzyme was added to 100 μl of stock DNA, this solution was incubated in a heat block (Thermomixer C, Eppendorf) at 37˚C for 15 mins. The enzyme was then deactivated at 65˚C for 20mins. Real-time PCRs were performed in 15 μl volumes including: 6.75 μl Jumpstart Taq Ready Mix (Sigma); 1.2 μl MgCl$_2$ (25mM); 0.45 μl of each primer; 0.15 μl probe; 4 μl of molecular grade water (Sigma); and 2 μl of template DNA. The standard real-time PCR cycle program was as follows. Hold stage: 50˚C for 2 mins then; 95˚C for 10 mins. PCR stage: 40 cycles of (95˚C for 15 secs; X˚C for 1 min), with primer annealing temperature X being 58, 60, 62, 64, or 68; depending on the experiment. Primer concentration, MgCl$_2$ concentration and temperature was adjusted for validation and optimization of the assay as described below. Reactions were performed on a "QuantStudio 6 Flex" (Applied Biosystems) real-time PCR machine and analysis was done on the "QuantStudio Real-Time PCR Software" (Applied Biosystems).

## 2.5. Assay validation

**2.5.1. Specificity.** The final primer and probe set was tested on genomic DNA from 47 *B. cockerelli* specimens from different life stages. These included the 4 US biotypes [17,54] and specimens from New Zealand to determine false negatives. The assay was tested for specificity against genomic DNA of 73 non-target psyllid species collected as mentioned above, to detect false positives. This included a total of 8 other closely related *Bactericera* spp. and the major

vectors of Lso on Apiaceous crops (*B. nigricornis*, *B. trigonica* and *Trioza apicalis*). Information regarding samples tested is in results section 3.1. The assay was also checked for cross-reaction against potato genomic DNA (*Solanum tuberosum*), 3 samples of *S. tuberosum* 'Maris Piper' were tested in replicates of 8. All reactions with non-target DNA were run in conjunction with a TaqMan Exogenous Internal Positive Control Reagent Kit (Applied Biosystems) to rule out the possibility that false positives were not obtained due to inhibition within the reaction. DNA from all non-target psyllids was sequenced in either ITS2, CO1 or both to ensure psyllid DNA was present in all reactions to rule out false negatives due to inefficient DNA extraction. Reactions were performed in duplicate at least, with a higher number of replicates for species closely related to *B. cockerelli*. False positives were defined as reactions with non-target DNA that showed fluorescence above the cycle threshold during 40 cycles; and false negatives were defined as reactions with *B. cockerelli* DNA that did not give a $C_t$ after 40 cycles.

**2.5.2. Sensitivity.** Experiments were performed to determine the limit of detection of the assays. DNA standards were produced using *B. cockerelli* linearized cloned DNA from the ITS2 region. A nine point 10-fold dilution series starting with 10 ng/μl DNA up to $10^{-8}$ ng/μl of linearised plasmid DNA and genomic DNA was used to determine the limit of detection. 100ng/μl stock DNA concentration was initially checked using QuBit 4 Fluorometer (Invitrogen) and 5 μl was added to 45 μl of molecular grade water (Sigma-Aldrich) to dilute 1:10; eight subsequent dilutions were made. Linearised and non-linearised DNA was compared along with genomic DNA. The ability of the assay to detect immatures and eggs was also tested. DNA from various instars of immatures was extracted using the non-destructive protocol described above. Batches of 1 egg, 5 eggs and 10 eggs were extracted using the DNeasy Blood & Tissue kit (Qiagen) and initially broken with a pestle.

**2.5.3. Repeatability and reproducibility.** Variation in the performance of the assay between runs and within runs was assessed at a 0.2 μM primer concentration, with 1.5mM $MgCl_2$, and 60˚C annealing temperature. Linearised plasmid DNA from *Escherichia coli* transformed with *B. cockerelli* ITS2 DNA was used. A six point 1:10 dilution series starting at 10ng/μl was used with each dilution being performed in triplicate. The same experiment was repeated 3x simultaneously. Runs and variations between the three experiments were recorded and analysed using QuantStudio 6 Real-Time PCR Software. An identical plate following the same plate set-up and reaction mix was run simultaneously on another QuantStudio 6 real-time PCR machine to compare inter-run variation.

**2.5.4. Robustness/optimization.** Amplification of target DNA, specificity and sensitivity at different $MgCl_2$ concentration, primer concentrations and annealing temperatures were performed to assess robustness. The assay was tested with 1.5, 3.5, 5.5, 7.5 and 9.5mM $MgCl_2$ concentration. For primers, 0.1, 0.2, 0.3, 0.5 and 1.0 μM concentrations were tested. The assay was also tested at different annealing temperatures 58, 60, 62, 64, 68˚C across. For each tested parameter, optimization was performed across a nine point 1:10 dilution series starting at 10ng/μl DNA. All samples were tested in triplicates. Closely related *Bactericera* species were included in these assays to assess specificity under different assay conditions. After optimization of the assay a multifactorial robustness test was performed across two different real-time PCR machines to test the combined effects of small changes/errors in the PCR set-up. The assays were run on a "QuantStudio 6 Flex" (Applied Biosystems) and "CFX96 Real-Time System" (BioRad); results were analysed using "QuantStudio 6 Real-Time PCR Software" (Applied Biosystems) and "CFX Manager 3.1" (BioRad). The methodology used followed the European Network of GMO Laboratories (ENGL) recommendations [55].

## 3. Results

### 3.1. DNA extraction, PCR, and DNA sequencing for identification of psyllids

DNA from 110 psyllid specimens comprising 73 different species were extracted, amplified and sequenced successfully from either CO1 or ITS2 gene regions, or both (Table 1).

### 3.2. Bioinformatics and real-time PCR assay design

While differentiation within both the ITS2 and CO1 gene regions was sufficient to discriminate between psyllid species, the ITS2 gene region was more suitable for TaqMan assay design for *B. cockerelli*. Similarities between CO1 gene sequences between members of the *Bactericera* genus and *B. cockerelli* were higher than in the ITS2 region (average % similarity = 82.51 ± 0.68 for CO1 and 77.80 ± 4.79 for ITS2) (Table 2). The ITS2 region showed larger sections of variability along the gene on which to design primers and probes. Several primer and probe sets passed the selection criteria, but most were unsuitable due to high rate of false positives from closely related *Bactericera* species. The final primer and probe set Bcoc_JSK2 (Table 3) targets a 187bp region of the ITS2 gene (Fig 1).

### 3.3. Specificity and sensitivity

This assay did not amplify DNA from any of the 73 non-target psyllid species or *Solanum tuberosum* DNA when tested at 60˚C with primer concentration 0.2 μM. Samples included nine closely related *Bactericera* species with similar ITS2 and CO1 sequences (Table 2). Under optimal conditions, false negatives = 0% for all non-target species tested with pure genomic DNA, giving a diagnostic specificity of 100%. Some suboptimal reaction conditions showed 33% false positives against high concentrations (10 ng / 1 ng) of *Bactericera albiventris* cloned DNA (see below). All *B. cockerelli* genomic DNA samples gave positive results (Table 4) giving 0% false negatives across 54 biological replicates and 147 technical replicates; resulting in a diagnostic sensitivity of 100%. These included *B. cockerelli* specimens from each of the four US biotypes as well as specimens from New Zealand. These specimens included adults, immature stages and eggs. The assay can amplify *B. cockerelli* DNA from both cloned and genomic samples. Under optimal conditions for PCR efficiency and specificity (60˚C, 0.2 μM primer, 1.5 mM MgCl$_2$) the limit of detection was 0.000001 ng DNA across a range of different reaction parameters this equates to 200 copy numbers of ITS2 calculated using the following equation: Number of Copies = (ng DNA x 6.022x10$^{23}$) ÷ (length of plasmid (4656) + cloned fragment (700)bp) * 1x10$^9$ * 660). The copy number calculator available at http://scienceprimer.com/copy-number-calculator-for-realtime-pcr was used. Diagnostic sensitivity was 100% on all DNA extracted from *B. cockerelli* immatures. False negatives from DNA from egg extractions were 0% for single eggs and 0% for batches of 3 and 10 eggs.

### 3.4. Repeatability and reproducibility

No significant differences were found between $C_t$ means across the different replicates at different concentrations as tested by two-way ANOVA ($F_{5, 25}$ = 0.54, $p$ = 0.955). The assay also performed consistently across different machines and there was no significant difference between runs across the two machines as tested by two-way ANOVA ($F_{1, 5}$ = 1.28, $p$ = 0.279).

**Table 1. Information on non-target psyllid species and plant specimens tested using the *B. cockerelli* real-time PCR assay Bcoc_JSK2 showing number of technical replicates and false positives.**

| Family | Genus | Species | Voucher ID | Collection Location | Collection method | CO1 Ac# | ITS2 Ac# | Tech Reps | Voucher Location |
|---|---|---|---|---|---|---|---|---|---|
| **Aphalaridae** | *Aphalara* | *avicularis* | 160718.A.avi.23 | Wellesbourne, UK | suction trap | MT021761 | / | 2 | 1 |
| | | *polygoni* | 160718.A.pol.22 | Wellesbourne, UK | suction trap | / | MT038907 | 2 | 1 |
| | *Blastopsylla* | *occidentalis* | 180312.Bl.occ.24 | Salamanca, Spain | suction trap | MN272146 | MN316692 | 2 | *3* |
| | *Craspedolepta* | *gutierreziae* | 160825.5 US | Nevada, USA | field collection | MT021786 | MT038962 | 2 | 1 |
| | | *minutissima* | 160825.1 US | Nevada, USA | field collection | MT021787 | MT038963 | 2 | 1 |
| | | | 160825.10 US | Oregon, USA | field collection | MT021789 | / | 2 | 1 |
| | | | 160825.4 US | Nevada, USA | field collection | MT021788 | / | 2 | 1 |
| | | *nervosa* | 160728.Cra.ner.2 | Gogarbank, UK | suction trap | MT021790 | MT038964 | 2 | 1 |
| | | *pinicola* | 160825.2 US | Nevada, USA | field collection | / | MT038965 | 2 | 1 |
| | | *subpunctata* | 160421.C.sub.5 | Gogarbank, UK | suction trap | MT021791 | MT038966 | 2 | 1 |
| | *Rhinocola* | *aceris* | 151014.R.ace.14 | Wellesbourne, UK | suction trap | MT021810 | MT038979 | 2 | 2 |
| **Liviidae** | *Diaphorina* | *citri* | 160309.D.cit.6 | Lab Colony, Vietnam | Lab Reared | MT021794 | MT038969 | 2 | 1 |
| | *Euphyllura* | *olivina* | 180125.Eup.oli.3 | imports from Italy | imported *Olea europeae* | MT021797 | MT038970 | 2 | *3* |
| | *Livia* | *crefeldensis* | 180312.L.cre.5 | Salamanca, Spain | suction trap | MN316678 | MN272127 | 2 | *3* |
| | | *junci* | 160404.L.jun.1 | Broom' s Barn, UK | suction trap | MT021801 | / | 2 | 2 |
| | | *opaqua* | 160825.6 US | Nevada, USA | field collection | MT021802 | MT038973 | 2 | 1 |
| **Psyllidae** | *Arytaina* | *genistae* | 151203.A.gen.2J | Ayr, UK | suction trap | / | MT038909 | 2 | 1 |
| | *Arytainilla* | *gredi* | 180312.A.gre.1 | Salamanca, Spain | suction trap | MN272123 | MN316677 | 2 | *3* |
| | | *spartiophila* | 180716.A.spa.29 | Edinburgh, UK | suction trap | MT021762 | MT038908 | 2 | *3* |
| | *Baeopelma* | *foersteri* | 151203.B.foe.1J | Ayr, UK | suction trap | / | MT038944 | 2 | 1 |
| | | *foersteri* | 160928.B.foe.2 | SASA, UK | suction trap | MT021776 | / | 2 | 1 |
| | *Cacopsylla* | *affinis* | 151203.C.aff.1 | Wye, UK | suction trap | MT021777 | MT038945 | 2 | 2 |
| | | *ambigua* | 160404.C.amb.4 | Wye, UK | suction trap | / | MT038946 | 2 | 2 |
| | | *ambigua* | 161024.C.amb.3 | Preston, UK | suction trap | / | MT038947 | 2 | 1 |
| | | *americana* | 160825.3 US | Nevada, USA | field collection | MT021778 | MT038948 | 2 | 1 |
| | | *brunneipennis* | 160309.C.bru.8 | Wye, UK | suction trap | / | MT038949 | 2 | 2 |
| | | *crataegi* | 160404.C.cra.3 | Broom' s Barn, UK | suction trap | MT021779 | MT038950 | 2 | 2 |
| | | *mali* | 180910.C.mal.30 | Elcho, UK | field collection | / | MT038951 | 2 | *3* |
| | | *melanoneura* | 160718.C.mel.6 | Kirton, UK | suction trap | / | MT038952 | 2 | *3* |
| | | *moscovita* | 190109.C.mos.1 | Germany | suction trap | / | / | 2 | *3* |
| | | *peregrina* | 161024.C.per.11 | Silwood Park, UK | suction trap | MT021780 | MT038953 | 2 | 1 |
| | | *pruni* | 160203.C.pru.18 | Wellesbourne, UK | suction trap | / | MT038954 | 2 | 2 |

(*Continued*)

**Table 1.** (*Continued*)

| Family | Genus | Species | Voucher ID | Collection Location | Collection method | CO1 Ac# | ITS2 Ac# | Tech Reps | Voucher Location |
|---|---|---|---|---|---|---|---|---|---|
| | | *pulchra* | 160718.C.pul.15 | Elgin, UK | suction trap | / | MT038955 | 2 | 1 |
| | | *pyricola* | 160203.C.pco.2 | Wye, UK | suction trap | MT021781 | MT038956 | 2 | 2 |
| | | *saliceti* | 161024.C.sal.7 | York, UK | suction trap | / | MT038958 | 2 | 1 |
| | | *sorbi* | 161024.C.sor.8 | Preston, UK | suction trap | MT021782 | MT038959 | 2 | 1 |
| | | *rhamnicola* | 151014.C.rha.8 | Wellesbourne, UK | suction trap | / | MT038957 | 2 | 2 |
| | | *ulmi* | 171011.C.ulm.13 | Germany | suction trap | MT021783 | MT038960 | 2 | *3* |
| | *Ceanothia* | *ceanothi* | 160825.9 US | Oregon, USA | field collection | MT021784 | / | 2 | 1 |
| | *Chamaepsylla* | *hartigii* | 160728.Ch.har.1 | Gogarbank, UK | suction trap | MT021785 | MT038961 | 2 | 1 |
| | *Euglyptoneura* | *fuscipennis* | 160825.7 US | Oregon, USA | field collection | MT021795 | / | 2 | 1 |
| | | *robusta* | 160825.8 US | Oregon, USA | field collection | MT021796 | / | 2 | 1 |
| | *Heteropsylla* | *texana* | 160825.11 US | Texas, USA | field collection | MT021798 | / | 2 | 1 |
| | *Psylla* | *alni* | 161019.P.aln.1 | Sweden | suction trap | MT021804 | / | 2 | 1 |
| | | *buxi* | 180622.P.bux.22 | Scotland, UK | suction trap | MT021806 | MT038976 | 2 | *3* |
| | | *betulae* | 161123.P.bet.20 | Jokioinen, Finland | suction trap | MT021805 | MT038975 | 2 | *3* |
| | *Psyllopsis* | *discrepans* | 151002.P.dis.8 | Sweden | suction trap | MT021807 | / | 2 | 1 |
| | | *fraxini* | 180716.P.fri.33 | Edinburgh, UK | suction trap | MT021808 | MT038977 | 2 | *3* |
| | | *fraxinicola* | 160203.P.fra.6 | Wellesbourne, UK | suction trap | MT021809 | MT038978 | 2 | 2 |
| | *Spanioneura* | *fonscolombii* | 180802.S.fon.29 | Edinburgh, UK | field collection | / | MT038980 | 2 | *3* |
| **Spondyliaspidae** | *Ctenarytaina* | *spatulata* | 160404.Ct.spa.6 | Wye, UK | suction trap | MT021792 | MT038967 | 2 | 2 |
| | | *spatulata* | 161024.Ct.spa.5 | Wye, UK | suction trap | MT021793 | MT038968 | 2 | 1 |
| **Triozidae** | *Bactericera* | *albiventris* | 171214.B.alb.11 | Jokioinen, Finland | suction trap | / | MT038910 | 5 | *3* |
| | | *curvatinervis* | 161123.B.cur.42 | Jokioinen, Finland | suction trap | / | MT038911 | 5 | *3* |
| | | *dorsalis* | 160803.B.dor.2 | Florida, USA | lab colony | MT021763 | MT038912 | 5 | *3* |
| | | *maculipennis* | 190604.B.mac.1 | Lab Colony, USA | Lab Reared | / | MT038913 | 2 | *3* |
| | | | 190604.B.mac.2 | Lab Colony, USA | Lab Reared | / | MT038914 | 2 | *3* |
| | | | 190604.B.mac.3 | Lab Colony, USA | Lab Reared | / | MT038915 | 2 | *3* |
| | | | 190604.B.mac.4 | Lab Colony, USA | Lab Reared | / | MT038916 | 2 | *3* |
| | | | 190604.B.mac.5 | Lab Colony, USA | Lab Reared | / | MT038917 | 2 | *3* |
| | | | 190604.B.mac.6 | Lab Colony, USA | Lab Reared | / | MT038918 | 2 | *3* |
| | | | 190604.B.mac.7 | Lab Colony, USA | Lab Reared | / | MT038919 | 2 | *3* |
| | | *nigricornis* | 170324.B.nig.18 | Spain | field collection | MT021764 | MT038920 | 5 | *3* |

(*Continued*)

**Table 1.** (Continued)

| Family | Genus | Species | Voucher ID | Collection Location | Collection method | CO1 Ac# | ITS2 Ac# | Tech Reps | Voucher Location |
|---|---|---|---|---|---|---|---|---|---|
| | | | 170324.B.nig.22 | Spain | field collection | MT021765 | MT038921 | 5 | *3* |
| | | *salicivora* | 190116.B.sal.1 | Elgin, UK | suction trap | / | / | 6 | *3* |
| | | *striola* | 161123.B.str.9 | Jokioinen, Finland | suction trap | / | MT038922 | | |
| | | *tremblayi* | 170731.B.tre.5 | Belgrade, Serbia | field collection | / | MT038923 | 5 | *3* |
| | | | 190604.B.tre.17 | Spain | Lab Colony | / | MT038924 | 2 | *3* |
| | | | 190604.B.tre.18 | Spain | Lab Colony | / | MT038925 | 2 | *3* |
| | | | 190604.B.tre.19 | Spain | Lab Colony | / | MT038926 | 2 | *3* |
| | | | 190604.B.tre.20 | Spain | Lab Colony | / | MT038927 | 2 | *3* |
| | | | 190604.B.tre.21 | Spain | Lab Colony | / | MT038928 | 2 | *3* |
| | | *trigonica* | 170629.B.tri.16 | Tunisia | field collection | MT021766 | MT038929 | 3 | *3* |
| | | | 170629.B.tri.17 | Tunisia | field collection | / | MT038930 | 3 | *3* |
| | | | 170629.B.tri.18 | Tunisia | field collection | MT021767 | MT038931 | 3 | *3* |
| | | | 181010.B.tri.17 | Spain | Lab Colony | MT021768 | MT038932 | 2 | *3* |
| | | | 181010.B.tri.18 | Spain | Lab Colony | MT021769 | MT038933 | 2 | *3* |
| | | | 181010.B.tri.19 | Spain | Lab Colony | / | MT038934 | 2 | *3* |
| | | | 181010.B.tri.20 | Spain | Lab Colony | MT021770 | MT038935 | 2 | *3* |
| | | | 181010.B.tri.21 | Spain | Lab Colony | / | MT038936 | 2 | *3* |
| | | | 190604.B.tri.23 | Spain | Lab Colony | MT021771 | MT038937 | 2 | *3* |
| | | | 190604.B.tri.24 | Spain | Lab Colony | / | MT038938 | 2 | *3* |
| | | | 190604.B.tri.25 | Spain | Lab Colony | MT021772 | MT038939 | 2 | *3* |
| | | | 190604.B.tri.26 | Spain | Lab Colony | MT021773 | MT038940 | 2 | *3* |
| | | | 190604.B.tri.27 | Spain | Lab Colony | MT021774 | MT038941 | 2 | *3* |
| | | | 190604.B.tri.28 | Spain | Lab Colony | / | MT038942 | 2 | *3* |
| | | | 190604.B.tri.29 | Spain | Lab Colony | MT021775 | MT038943 | 2 | *3* |
| | *Heterotrioza* | *chenopodii* | 160203.H.che.11 | Kirton, UK | suction trap | / | MT038971 | 2 | 2 |
| | | | 160825.12 US | Washington, USA | field collection | MT021799 | / | 2 | 1 |
| | *Lauritrioza* | *alacris* | 160816.L.ala.2 | Spain | suction trap | MT021800 | MT038972 | 2 | 1 |
| | *Powellia* | *vitreoradiata* | 161024.P.vit.10 | Kirton, UK | suction trap | MT021803 | MT038974 | 2 | 1 |
| | *Trioza* | *albifrons* | 160825.18.US | Nevada, USA | field collection | MT021811 | MT038981 | 2 | 1 |
| | | *anthrisci* | 150708.T.ant.11 | Jokioinen, Finland | field collection | MT021812 | / | 2 | *3* |
| | | *apicalis* | 161019.T.api.5 | Sweden | field collection | MT021813 | / | 2 | *3* |
| | | *buxtoni* | 170324.T.bux.11 | Israel | field collection | MT021814 | MT038982 | 2 | *3* |
| | | *centranthi* | 161024.T.cen.9 | Wye, UK | suction trap | MT021815 | / | 2 | 1 |
| | | *cerastii* | 171214.T.cer.32 | Vikki, Finland | suction trap | MT021816 | MT038983 | 2 | *3* |
| | | *dispar* | 160718.T.dis.26 | Hellfreda, Sweden | suction trap | MT021817 | / | 2 | 1 |
| | | *erytreae* | 160808.ICA.19 | Spain | Lab Colony | / | MT038984 | 2 | 1 |
| | | *flavipennis* | 160421.T.fla.3 | Sweden | suction trap | MT021818 | MT038985 | 2 | 1 |

(*Continued*)

**Table 1.** (Continued)

| Family | Genus | Species | Voucher ID | Collection Location | Collection method | CO1 Ac# | ITS2 Ac# | Tech Reps | Voucher Location |
|--------|-------|---------|-----------|---------------------|-------------------|---------|----------|-----------|------------------|
| | | *galii* | 160203.T.gal.23 | Wellesbourne, UK | suction trap | / | MT038986 | 2 | 2 |
| | | *remota* | 160718.T.rem.8 | Sweden | suction trap | / | MT038987 | 2 | 1 |
| | | | 180424.T.rem.1 | Dundee, UK | Suction trap | MT021819 | MT038988 | 3 | *3* |
| | | | 180424.T.rem.6 | Dundee, UK | Suction trap | MT021820 | MT038989 | 3 | *3* |
| | | | 180424.T.rem.16 | Dundee, UK | Suction trap | MT021821 | MT038990 | 3 | *3* |
| | | | 180424.T.rem.18 | Dundee, UK | Suction trap | MT021822 | MT038991 | 3 | *3* |
| | | | 180424.T.rem.19 | Dundee, UK | Suction trap | / | MT038992 | 3 | *3* |
| | | | 190116.T.rem.7 | UK | Suction trap | MT021823 | MT038993 | 3 | *3* |
| | | *rhamni* | 151002.T.rha.13 | Sweden | suction trap | MT021824 | MT038994 | 2 | 1 |
| | | *tatrensis* | 160718.T.tat.27 | Sweden | suction trap | / | MT038995 | 2 | 1 |
| | | *urticae* | 160816.T.urt.17 | Spain | field collection | / | MT038996 | 2 | 1 |

All non-target species gave 0% false positives. GenBank Accession numbers are included for ITS2 and CO1 regions if sequencing was successful. Voucher Location: 1 = 1; 2 = 2 Research Insect Survey; 3 = SASA Hemipteran DNA Database. All DNA samples are stored in the SASA Hemipteran DNA database. "/" = no sequence obtained.

**Table 2. Closely related *Bactericera* species tested with Bcoc_JSK2 assay.**

| Species | ITS2 | | | CO1 | | |
|---------|------|----|----|-----|----|----|
| | % Similarity | bp | GC content % | % Similarity | bp | GC content % |
| *B. trigonica* | 78.96 | 662 | 59.3 | 82.88 | 509 | 35.4 |
| *B. tremblayi* | 79.16 | 665 | 59.1 | 82.97 | 682 | 33 |
| *B. curvatinervis* | 80.30 | 655 | 58 | 82.23 | 678 | 34.7 |
| *B. nigricornis* | 81.16 | 668 | 59.3 | 81.28 | 521 | 36.7 |
| *B. albiventris* | 76.67 | 667 | 59.2 | 83.41 | 663 | 32.9 |
| *B. dorsalis* | 65.59 | 560 | 61.3 | 82.31 | 685 | 32.6 |
| *B. maculipennis* | 80.67 | 674 | 61.6 | nd | nd | nd |
| *B. salicivora* | nd | nd | nd | nd | nd | nd |
| *B. striola* | 79.91 | 663 | 59.1 | nd | nd | nd |
| *B.cockerelli* | N/A | 569 | 61.0 | N/A | 595 | 32.6 |

ITS similarity = % identity to DNA sample 150727.B.coc.02. CO1 similarity = % identity to a consensus sequences of all *B. cockerelli* sequences obtained during this study. CO1 genes showed higher similarity and fewer variable regions compared to ITS2 regions. Highest % similarity to *B. cockerelli* in the ITS2 region was found in *B. nigricornis* (81.16) and to *B. albiventris* in the CO1 region (83.41). The Bcoc_JSK2 assay does not give false positives with any of the species listed here. (nd = not determined due to sequencing failing).

**Table 3. Final oligonucleotide sequences for the Bcoc_JSK2 TaqMan real-time PCR assay to identify *B. cockerelli*.** The assay targets a 187 bp region of the ITS2 gene region.

| Oligo Name | Function | Sequence 5'-3' | Tm | Length (bp) |
|---|---|---|---|---|
| Bcoc_JSK2-f | forward primer | GAGGTCTCCTCATCGTGCGT | 61 | 25 |
| Bcoc_JSK2-r | reverse primer | GGACGAGCATTGCTGCTGC | 62.2 | 23 |
| Bcoc_JSK2-p | probe (FAM-BHQ) | GCAAACGCGGCACAAGTACCGCGC | 70.9 | 25 |

## 3.5. Robustness/optimization

The assays amplified *B. cockerelli* DNA at all primer concentrations, MgCl$_2$ concentrations and annealing temperatures with varying levels of efficiency, precision, and sensitivity (S1–S3 Tables). At primer concentration 0.5 μM, the assay was less sensitive only amplifying down to 0.0001 ng DNA. At higher primer concentrations (1.0 μM,) the assay showed higher sensitivity, but efficiency was outside the range for acceptable use. The assay performed optimally at 0.2 μM primer concentration showing good efficiency and high sensitivity (0.000001 ng DNA) (S1 Table). Generally, standard deviation of the C$_t$ was lower at higher DNA concentrations and some of the primer concentrations showed SD slightly above the accepted level for

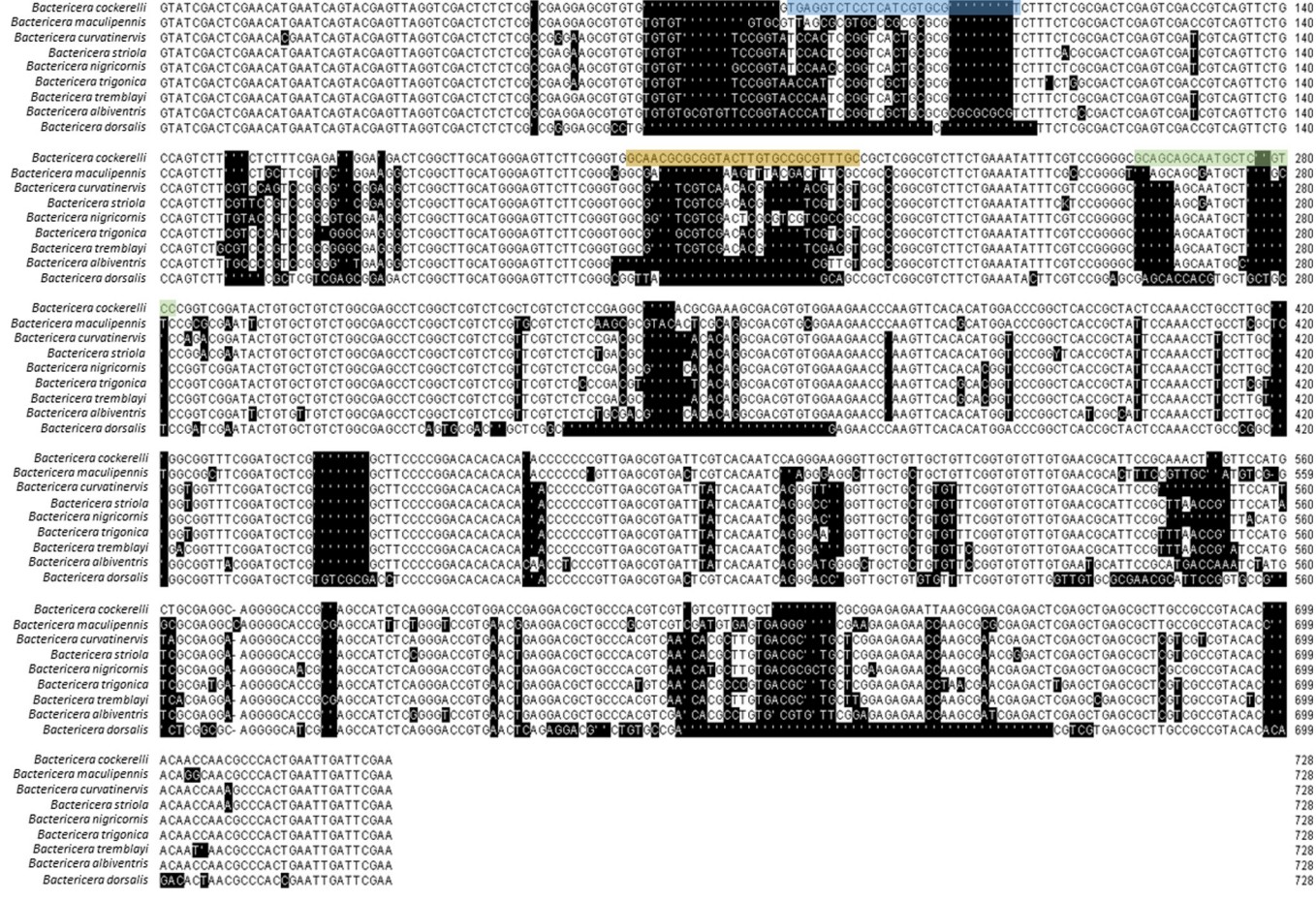

**Fig 1. CLUSTAL-W alignment of ITS2 regions from closely related *Bactericera* species showing variable regions and the gene target for the Bcoc_JSK2 primer and probe set.** Bases shades with black show differences to *B. cockerelli* sequence. Colour highlights locations of forward primer (blue highlight); reverse primer (green highlight) and probe (yellow highlight). The probe and reverse primer are reverse compliments of the highlighted regions here.

**Table 4. Information on *Bactericera cockerelli* samples tested with Bcoc_JSK2 assay including genomic DNA from adults, immatures, single eggs and egg batches.** Location of samples collection is also included. All samples gave 100% positives. Accession numbers for CO1 and ITS2 (MT027551-MT027599) regions are included. "/" = no sequence obtained.

| Sample name | Life Stage | Origin | $C_t$ ave | Tech reps | CO1 Ac# | ITS2 Ac# | DNA Source |
|---|---|---|---|---|---|---|---|
| 181119.B.coc.06 | 1 egg | Mexico | 29.80 | 2 | / | MT027568 | Genomic |
| 191003.B.coc.01 | 1 egg | Mexico | 33.41 | 3 | / | MT027592 | Genomic |
| 191003.B.coc.02 | 1 egg | Mexico | 24.95 | 3 | / | MT027593 | Genomic |
| 191003.B.coc.03 | 1 egg | Mexico | 33.79 | 3 | / | MT027594 | Genomic |
| 191003.B.coc.04 | 1 egg | Mexico | 22.43 | 6 | / | MT027595 | Genomic |
| 181119.B.coc.07 | 5 eggs | Mexico | 24.42 | 2 | / | MT027569 | Genomic |
| 181119.B.coc.21 | 5 eggs | Mexico | 28.32 | 2 | / | MT027582 | Genomic |
| 181119.B.coc.08 | 10 eggs | Mexico | 29.61 | 2 | / | MT027570 | Genomic |
| 181119.B.coc.22 | 10 eggs | Mexico | 26.43 | 2 | / | MT027583 | Genomic |
| 181119.B.coc.03 | immature | Mexico | 22.56 | 2 | / | MT027565 | Genomic |
| 181119.B.coc.04 | immature | Mexico | 22.33 | 2 | / | MT027566 | Genomic |
| 181119.B.coc.05 | immature | Mexico | 21.46 | 2 | / | MT027567 | Genomic |
| 181119.B.coc.11 | immature | Mexico | 23.16 | 2 | / | MT027573 | Genomic |
| 181119.B.coc.12 | immature | Mexico | 24.15 | 2 | / | MT027574 | Genomic |
| 181119.B.coc.13 | immature | Mexico | 23.94 | 2 | / | MT027575 | Genomic |
| 181119.B.coc.14 | immature | Mexico | 25.75 | 2 | / | MT027576 | Genomic |
| 181119.B.coc.16 | immature | Mexico | 23.49 | 2 | / | MT027578 | Genomic |
| 181119.B.coc.18 | immature | Mexico | 22.45 | 2 | / | MT027580 | Genomic |
| 181119.B.coc.19 | immature | Mexico | 23.50 | 2 | / | MT027581 | Genomic |
| 190604.B.coc.13 | immature | Mexico | 24.96 | 2 | / | MT027588 | Genomic |
| 190604.B.coc.14 | immature | Mexico | 25.09 | 2 | / | MT027589 | Genomic |
| 190604.B.coc.15 | immature | Mexico | 28.37 | 2 | / | MT027590 | Genomic |
| 150727.B.coc.02 | Adult | South Western, USA | 22.18 | 2 | MT040955 | MG719775 | Genomic |
| 150827.B.coc.02 | Adult | South Western, USA | 22.18 | 2 | MT040956 | MT027597 | Genomic |
| 150827.B.coc.03 | Adult | Central USA | 24.49 | 6 | MT040957 | MT027598 | Genomic |
| 150827.B.coc.04 | Adult | North Western, USA | 24.77 | 2 | MT040958 | MT027599 | Genomic |
| 150827.B.coc.06 | Adult | North Western, USA | 23.68 | 2 | MT040960 | MT027552 | Genomic |
| 150827.B.coc.12 | Adult | Western, USA | 20.39 | 2 | MT040961 | MT027596 | Genomic |
| 150827.B.coc.17 | Adult | South Western, USA | 19.65 | 2 | MT040962 | MT027553 | Genomic |
| 160725.B.coc.05 | Adult | Central, USA | 21.45 | 2 | MT040963 | / | Genomic |
| 160726.B.coc.01 | Adult | New Zealand | 21.56 | 2 | / | MT027557 | Genomic |
| 160726.B.coc.02 | Adult | New Zealand | 21.02 | 2 | / | MT027558 | Genomic |
| 160726.B.coc.03 | Adult | New Zealand | 20.48 | 2 | / | MT027559 | Genomic |
| 160726.B.coc.04 | Adult | New Zealand | 21.98 | 2 | / | MT027560 | Genomic |
| 160726.B.coc.05 | Adult | New Zealand | 19.43 | 2 | / | MT027561 | Genomic |
| 160726.B.coc.06 | Adult | New Zealand | 20.96 | 2 | / | MT027562 | Genomic |
| 180731.B.coc.04 | Adult | North Western, USA | 24.42 | 6 | MT040964 | / | Genomic |
| 180731.B.coc.05 | Adult | Western, USA | 22.91 | 6 | MT040965 | / | Genomic |
| 180731.B.coc.06 | Adult | Western, USA | 27.14 | 6 | MT040966 | / | Genomic |
| 181119.B.coc.01 | Adult | Mexico | 21.47 | 2 | / | MT027563 | Genomic |
| 181119.B.coc.02 | Adult | Mexico | 19.98 | 2 | / | MT027564 | Genomic |
| 181119.B.coc.09 | Adult | Mexico | 21.83 | 2 | / | MT027571 | Genomic |
| 181119.B.coc.10 | Adult | Mexico | 19.48 | 2 | / | MT027572 | Genomic |
| 181119.B.coc.15 | Adult | Mexico | 21.27 | 2 | / | MT027577 | Genomic |
| 181119.B.coc.17 | Adult | Mexico | 23.74 | 2 | / | MT027579 | Genomic |
| 190604.B.coc.09 | Adult | USDA, Lab Colony | 21.51 | 2 | / | MT027584 | Genomic |

(*Continued*)

**Table 4.** (Continued)

| Sample name | Life Stage | Origin | $C_t$ ave | Tech reps | CO1 Ac# | ITS2 Ac# | DNA Source |
|---|---|---|---|---|---|---|---|
| 190604.B.coc.10 | Adult | Mexico | 20.33 | 2 | / | MT027585 | Genomic |
| 190604.B.coc.11 | Adult | Mexico | 22.67 | 2 | / | MT027586 | Genomic |
| 190604.B.coc.12 | Adult | Mexico | 24.37 | 2 | / | MT027587 | Genomic |
| 190604.B.coc.16 | Adult | Mexico | 27.15 | 2 | / | MT027591 | Genomic |
| 150827.B.coc.05.col.04 | transformed *E. coli* | Lab | 11.23 | 6 | MT040959 | MT027551 | Cloned, 10ng |
| 160725.B.coc.01.col.06 | transformed *E. coli* | Lab | 11.55 | 6 | / | MT027554 | Cloned, 10ng |
| 160725.B.coc.06.col.04 | transformed *E. coli* | Lab | 11.78 | 6 | / | MT027555 | Cloned, 10ng |
| 160725.B.coc.07.col.08 | transformed *E. coli* | Lab | 11.67 | 6 | / | MT027556 | Cloned, 10ng |

quantitative real-time PCR, however this module is intended for qualitative use. At high DNA concentrations all primer concentrations are suitable for use with Bcoc_JSK2 primer and probe set to detect *B. cockerelli* but 0.2 μM is recommended for best results. The assay did not amplify non-target DNA from the 8 other *Bactericera* species tested at the different primer concentrations (0.1, 0.2, 0.3, 0.5 and 1.0 μM).

The MgCl2 concentration of the assay made only small differences to the overall performance of the assay (S2 Table) and the assay was able to amplify *B. cockerelli* DNA at low concentrations (0.000001 ng) at each $MgCl_2$ concentration. The precision of the assay was lower at higher $MgCl_2$ concentrations 7.5mM and 9.5mM (S2 Table).

Sensitivity was slightly higher at 64˚C giving 33.33% (n = 3) positives for only 20 copies of *B. cockerelli* DNA (0.0000001 ng), however at 64˚C and 66˚C 33.33% (n = 3) false positives were found with 10ng and 1 ng of *B. albiventris* cloned DNA (S3 Table). Reactions at 58˚C were 10 to 100-fold less sensitive than reactions at 64˚C. For best sensitivity and specificity, it is suggested that assays using the Bcoc_JSK2 primer and probe set should be performed at 60˚C or 62˚C. While higher temperatures appear to be more sensitive, they are not recommended on unknown samples due to the small likelihood of returning false positives with *B. albiventris* and possibly other un-tested *Bactericera* spp.

It is recommended that this assay be performed at 60˚C– 62˚C, with a $MgCl_2$ concentration of 1.5mM and a primer concentration of 0.2 μM. To test the robustness of these conditions a multifactorial approach was taken [55]. The assay performed satisfactorily across the different treatments and was shown to be robust and unaffected by small changes in assay set-up (S4 Table). Each treatment gave 100% positives for amplification of *B. cockerelli* genomic DNA.

## 4. Discussion

The Tomato-Potato psyllid is an economically damaging pest of solanaceous plants that has spread by human mediated dispersal. It causes feeding damage to plants but also is the major vector of '*Candidatus* Liberibacter solanacearum' (Lso), a phloem limited bacterium that is associated with disease in solanaceous and apiaceous plants. Management of this insect pest requires accurate identification of *B. cockerelli*, this is often difficult if eggs or immature life stages only are available for identification. Hitherto, identification of *B. cockerelli* required either considerable expertise in psyllid taxonomy or the lengthy process of DNA barcoding [54].

We have designed and validated the first species-specific, quantitative real-time PCR Taq-Man assay for *B. cockerelli* by using the comparison of 73 non-target species to identify unique gene regions that were suitable for primer/probe design and species differentiation. The genus *Bactericera* currently contains 160 species [20] and <1% of these have been tested in the current study due to the difficulty in obtaining other specimens from the field or lab colonies.

However Europe is home to 26 different species of *Bactericera* [20], 30% of which have been tested for false positives using this assay. Psyllid species that were tested are most commonly found in potato and carrot fields in Europe and the wider EPPO region which should minimize the potential for false positives and ensure the assay is efficient at detecting outbreaks in European fields. The assay was also tested on nine closely related *Bactericera* species. The number of species used in our study is relatively high compared to other reported TaqMan assays for plant pests that report lower numbers of non-target species [56,57].

The assay is based on a 187 bp region of the ITS2 gene which was suitable as it contained high interspecific variation consisting of stretches of insertions and deletions (INDELs). The ITS2 region has been used to distinguish species phylogenetically and to identify cryptic species in the *Cacopsylla pruni* complex [47]. DNA sequences obtained from this study will improve psyllid representation on online DNA databases, reducing the chance of Type II errors (i.e. misidentification due to lack of conspecific references) [58]. The *B. cockerelli* sequences on which we tested this assay (and many of the non-target psyllid species) were from different geographic locations to account for intraspecific variation. *Bactericera cockerelli* specimens from the four USA biotypes and specimens from New Zealand all gave 100% true positives.

The success rates of eradications are dependent on the length of time between introduction, detection, and implementation of eradication measures as Lso displays a short transmission time from *B. cockerelli* to potatoes [4,25]. Feasibly, methodology described in this study could be used to extract DNA from a specimen and test for *B. cockerelli* positives within 6–12 hrs or quicker. This is faster than identification by DNA barcoding and could aid in eradications/prevention of incursions. This time could be reduced further if the real-time assay is used in conjunction with faster DNA extraction protocols.

There are currently no methods described within the EPPO "agreed diagnostic protocol for identification of *B. cockerelli*" [4]. In addition, the current EPPO control system for *B. cockerelli* and Lso [4] highlights the importance of identifying psyllid eggs and immatures on various plant materials during inspections and monitoring but gives minimal guidelines for achieving this. Validation of this assay demonstrates that it would be a reliable and accurate tool for use in this area and it will therefore be prepared for consideration by the EPPO diagnostic panel. This assay is also useful for monitoring *B. cockerelli* occurrence at several spatial scales, from local border checks to regional surveys which use different trapping methods (water, sticky, suction, aerial balloon traps) where no host plant data is available. Given the sensitivity of this assay it should be possible to detect *B. cockerelli* DNA from insect fragments (e.g. legs, heads) if DNA extraction is adequate. However, further validation should be performed to ensure the assay performs adequately on samples obtained from different traps. This assay should be tested on additional congeneric species and other closely related Triozidae psyllids. Another limitation of this assay is that it cannot yet be taken out into the field, making it less portable than LAMP assays or other NGS sequencing techniques such as Nanopore technology.

In conclusion a rapid, specific, robust, repeatable and reliable real-time PCR assay has now been validated and can be used to detect the important pest *B. cockerelli*. This will be an important tool for providing much-needed support to prevent new outbreaks. The assay can be implemented by practitioners with molecular biology experience and does not require personnel to have classical taxonomic knowledge of insects or psyllids; making this tool more accessible than traditional methods. The assay can be used to complement field surveillance and may facilitate further ecological studies of *B. cockerelli* requiring the identification of immatures and eggs. The strength of this assay lies in the collaboration of molecular biologists and classical taxonomists working together to build a reliable database for DNA barcoding of psyllids.

## Supporting information

**S1 Table. Assay performance across a range of primer concentrations at 60˚C and 1.5mM MgCl₂.** Optimum primer concentration was 0.2 μM showing the best combination of $r^2$, slope, efficiency, and sensitivity.
(DOCX)

**S2 Table. Performance of *B. cockerelli* real-time PCR assay at different magnesium chloride (MgCl₂) concentrations.**
(DOCX)

**S3 Table. Summary of standard curves from optimisation of temperature on Bcoc_JSK2 real-time PCR assay for identification of B. cockerelli.** All DNA concentrations tested above the limit of detection (10ng, 1 ng, 0.1ng, 0.01ng 0.001ng, 0.0001ng, 0.00001ng, 0.000001ng) gave 100% positives across 3 x replicates. LOD is given for each temperature. All non-target Bactericera species tested at different DNA concentration gave 0% false positives except for B. albiventris cloned DNA which cross reacted at 64 and 66˚C. (*reactions at 64˚C gave 33.33% positives at 20 copy numbers).
(DOCX)

**S4 Table. Set-up and results of multifactorial robustness experiment testing the Bcoc_JSK2 assay on *B. cockerelli* genomic DNA.** All treatments showed 100% positives despite small changes to the overall set-up.
(DOCX)

## Acknowledgments

We thank A. Fereres & C. A. Antolínez Delgado (Institute of Agricultural Sciences, CSIC, Spain), A. Nissinen (Natural Resources Institute Finland), J. Munyaneza, R. Cooper, M. Heidt, K. Swisher Grimm (USDA Agricultural Research Services), S. Bulman (Plant and Food Research, New Zealand), A. Jensen, S. Halbert (Florida Department of Agriculture & Consumer Services, Dept. of Plant Industry) and Alberto Flores (Universidad Autónoma Agraria Antonio Narro) for specimens; and thank C. Jeffries, L. Webster, V. Mulholland, and A. Reid (SASA) for providing advice. We also thank SASA Potato Genotyping team for providing Potato DNA.

## Author Contributions

**Conceptualization:** J. C. Sumner-Kalkun, M. J. Sjölund, F. Highet, J. R. Bell, D. M. Kenyon.

**Data curation:** J. C. Sumner-Kalkun, M. J. Sjölund, Y. M. Arnsdorf, M. Carnegie, F. Highet, D. Ouvrard, A. F. C. Greenslade, J. R. Bell, R. Sigvald.

**Formal analysis:** J. C. Sumner-Kalkun, M. J. Sjölund, Y. M. Arnsdorf, M. Carnegie, D. Ouvrard, A. F. C. Greenslade.

**Funding acquisition:** J. R. Bell, D. M. Kenyon.

**Investigation:** J. C. Sumner-Kalkun, M. J. Sjölund, Y. M. Arnsdorf, D. Ouvrard, A. F. C. Greenslade.

**Methodology:** J. C. Sumner-Kalkun, M. J. Sjölund, F. Highet, J. R. Bell, D. M. Kenyon.

**Project administration:** J. C. Sumner-Kalkun, M. J. Sjölund, F. Highet, J. R. Bell, D. M. Kenyon.

**Resources:** M. J. Sjölund, M. Carnegie, D. Ouvrard, A. F. C. Greenslade, J. R. Bell, R. Sigvald, D. M. Kenyon.

**Software:** J. C. Sumner-Kalkun.

**Supervision:** J. C. Sumner-Kalkun, M. J. Sjölund, F. Highet, J. R. Bell, D. M. Kenyon.

**Validation:** J. C. Sumner-Kalkun, M. J. Sjölund, Y. M. Arnsdorf.

**Visualization:** J. C. Sumner-Kalkun, M. J. Sjölund.

**Writing – original draft:** J. C. Sumner-Kalkun, M. J. Sjölund, F. Highet.

**Writing – review & editing:** J. C. Sumner-Kalkun, M. J. Sjölund, F. Highet, D. Ouvrard, A. F. C. Greenslade, J. R. Bell.

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
