## [Decision Letter · Decision Letter 0]

15 Jan 2020

PONE-D-19-35047

A diagnostic real-time PCR assay for the rapid identification of the tomato-potato psyllid, Bactericera cockerelli (Šulc, 1909) and development of a psyllid barcoding database.

PLOS ONE

Dear Dr. Sumner-Kalkun,

Thank you for submitting your manuscript to PLOS ONE. After careful consideration, we feel that it has merit but does not fully meet PLOS ONE’s publication criteria as it currently stands. Therefore, we invite you to submit a revised version of the manuscript that addresses the points raised during the review process.

This manuscript fell in a grey area between minor and major revisions. Three different reviewers examined the manuscript, and I also reviewed it. I agree with the 1st reviewer that you may be stretching a bit and could possibly focus some. This work will provide a useful tool. I think that alone makes it worth publication, and that opinion is shared by the reviewers. I also think that it is a complete and comprehensive piece of work. I, therefore, encourage you to focus on the comments form reviewer 1 and those about length etc. when preparing a resubmission.

We would appreciate receiving your revised manuscript by Feb 29 2020 11:59PM. To enhance the reproducibility of your results, we recommend that if applicable you deposit your laboratory protocols in protocols.io, where a protocol can be assigned its own identifier (DOI) such that it can be cited independently in the future. For instructions see: http://journals.plos.org/plosone/s/submission-guidelines#loc-laboratory-protocols

We look forward to receiving your revised manuscript.

Kind regards,

Sean Michael Prager, Ph.D.

Academic Editor

PLOS ONE

Journal Requirements:

Reviewers' comments:

Reviewer's Responses to Questions

**Comments to the Author**

1. Is the manuscript technically sound, and do the data support the conclusions?

Reviewer #1: Yes

Reviewer #2: Yes

Reviewer #3: Yes

2. Has the statistical analysis been performed appropriately and rigorously? 

Reviewer #1: Yes

Reviewer #2: Yes

Reviewer #3: Yes

3. Have the authors made all data underlying the findings in their manuscript fully available?

Reviewer #1: No

Reviewer #2: No

Reviewer #3: No

4. Is the manuscript presented in an intelligible fashion and written in standard English?

Reviewer #1: Yes

Reviewer #2: Yes

Reviewer #3: Yes

5. Review Comments to the Author

Reviewer #1: This manuscript describes a qPCR assay to identify potato psyllid intercepted in shipments. The assay is paramount to Europe's ability to detect potential introductions of this psyllid, which would be harmful to agricultural production. The authors describe the assay and confirmed that it does not amplify the ITS gene of other psyllids.

My major concern for the manuscript is that it is overwritten and over-interpreted. The study is very simple - qPCR assay to detect potato psyllid - yet the text is over 50 pages long, includes unrelated information in the introduction, and includes an overly long discussion. The manuscript should be re-written to focus only on the assay and its use in trade commodities. Specific comments are provided in an attached document. I will apologize for my handwriting.

Reviewer #2: In this manuscript, the authors describe the design and validation of the first species-specific TaqMan probe-based real-time PCR assay, targeting the ITS2 gene region of Bactericera cockerelli, for robust and quick identification of the potato-tomato psyllid B. cockerelli, the main vector of ‘Candidatus Liberibacter solanacearum’ on potato and tomato crops in Central and Northern America and New Zealand. The authors examined false-positive rates in non-target psyllid species and false-negative rates in target species, including B. cockerelli at different life stages. The assay also compared amplification efficiency at different MgCl2 concentrations, primer concentrations, and annealing temperatures, and determined the detection limit at optimum conditions. The assay was designed and presented in a very robust way, however, I have some minor concerns the authors need to look into before the manuscript can be accepted.

Minor Concerns:

1. Data Availability: The authors need to add accession numbers for their sequence data.

2. Page 8 Line 163: What part of the body is used for micro-dissection to extract DNA? The authors should describe the micro-dissection procedure in more detail rather than only citing the papers.

3. Page 8 Line 172: “For amplification of ITS2 primers CA55p8sFcm-F and CA28sB1d-R [60] and for amplification of CO1 gene regions arthropod barcoding Primers LCO1490 and HCO2198 [61].” The authors should check the grammar here. It is not a complete sentence. It could be “For amplification of ITS2, primers CA55p8sFcm-F and CA28sB1d-R [60] were used, and for amplification of CO1 gene regions, arthropod barcoding Primers LCO1490 and HCO2198 [61] were used.”

4. Page 10 Line 204: “DNA was extracted as above using the non-destructive method, amplified and cloned into competent Escherichia coli cells using the TOPO TA cloning kit (Thermo-Fisher).” The authors should specify what genes (ITS2 or CO1?) they amplified for cloning, and what restriction enzyme (EcoRI?) they used to linearize the plasmid.

5. Page 10 Line 212: The authors need to list the real time PCR cycling conditions here, for example XX degrees for XX seconds.

6. Page 10 Line 223: “All reactions with non-target psyllid DNA were run in conjunction with a TaqMan Exogenous Internal Positive Control Reagent Kit (Applied Biosystems) to ensure false positives were not obtained due to inhibition within the reaction”. Here, “ensure” should be “rule out the possibility that”.

7. Page 11 Line 226: “DNA from all non-target psyllids was sequenced to ensure psyllid DNA was present in all reactions to rule out false negatives due to inefficient DNA extraction.” What DNA was sequenced? PCR product from ITS2 or CO1? The authors need to specify.

8. Page 11 Line 239: “6 subsequent dilutions were made. Stock DNA 10 ng/μl was linearised using EcoRI restrictions enzyme (New England Biolabs),” Here “6 subsequent dilutions” should be “8 subsequent dilutions”, according to the nine point 10-fold dilution series mentioned on Page 11 Line 236.

9. Page 12 Line 252: “A six point 1:10 dilution series starting at 10ng/μl was used with each dilution being performed in triplicate.” Here, “six point” should be “nine point” according to Page 11 Line 236.

10. Page 12 Line 263: “For each tested parameter, optimization was performed across an eight point 1:10 dilution series starting at 10ng DNA.” Here, “eight point” should be “nine point”, “10ng” should be “10ng/μl”.

11. In Supplementary table S1, green and red color coding should be explained in the text. What does TBC mean? Accession numbers should be given for all the sequences. Accession numbers in Table 3 should also be given and TBC should be explained.

12. Page 14 Line 289: “CO1 genes showed higher similarity and generally less conserved and variable regions compared to ITS2 regions.” Here “less conserved and variable” should be “less variable”.

13. Page 17 Line 310: “0.2 µ/mol” should be “0.2 µM”.

14. Page 18 Line 324: “The copy number calculator available at http://scienceprimer.com/copy-number-calculator-for-realtime-pcr was used.” Here a hyperlink should be created. According to the link and the formula given, 0.00001ng DNA equals 4.879×10000 copies, if length of gene region is considered 187bp (product length of ITS2 in real time PCR). However, the authors calculated that it equals to 200bp. Please double check the calculation.

15. Page 18 Line 337: “At primer concentration, 0.5 μM the assay was less sensitive only amplifying up to 0.001 ng DNA.” It should be “At primer concentration 0.5 μM, the assay was less sensitive only amplifying up to 0.001 ng DNA.”

16. Page 18 Line 338: “At higher primer concentrations (0.5 and 1.0) the assay showed higher sensitivity” Here “(0.5 and 1.0)” should be “(1.0 μM)”.

17. Page 19 Line 350: “The precision of the assay was lower at higher MgCl2 concentrations 6mM and 8mM (Supp Tab. S3).” Here “6mM and 8mM” should be “7.5mM and 9mM”.

18. Page 19 Line 354: “Reactions at 58 °C were 10 to 100-fold less sensitive than reactions at 58 °C.” Here it should be “Reactions at 58 °C were 10 to 100-fold less sensitive than reactions at 64 °C.”

19. Page 20 Line 367: “We have designed and validated the first species-specific, qualitative real-time PCR TaqMan assay for B. cockerelli by using the comparison of 73 non-target species to identify unique gene regions that were suitable for primer/probe design and species differentiation.” Here “qualitative” should be “quantitative”.

Reviewer #3: The manuscript presents a new real time assay that will make identification of the key pest commonly known as potato-tomato psyllid easier and faster. The assay has been rigorously developed, with appropriate controls, replication and sample size. The specificity of the assay is fairly assured by the inclusion, in its development and validation stage, of numerous non target species, including 9 congeneric species, representing about 30% of known European Bactericera taxa. The manuscript is well written, with thorough introduction and discussion, and methods and results clearly presented. There are only some minor issues that should be dealt with before the manuscript can be accepted for publication:

- Page 8 line 175: please replace amount of primers used with final concentration of primers (or add this)

- Page 10 line 213: please add cycling conditions of real time PCR, as done for CO1 and ITS2 amplification

- Table 1: should include also B. cockerelli, so to include fragment size of amplicons for this species. In alternative, fragment sizes can be added to the main text

Table 3: not clear what the "/" symbol in the CO1 column means

- Page 17 line 310: please check spelling of concentration

- Page 17 line 316: numbers seem not to add up: how many technical replicates were used per sample?

- Page 18 line 323: I have tried the formula myself using the concentration (0.00001 ng) and fragment size (187 bp) specified by the authors, but I get a quite different number of ITS2 copies (about 50,000 versus 200). Please double check, and add actual numbers to the formula.

Of some concern is the author's answer to the data accessibility question. Authors stated that they are not going to make all data available, with a generic "Some restrictions will apply". Please explain what data will not be made accessible and why.

6. PLOS authors have the option to publish the peer review history of their article (what does this mean?). If published, this will include your full peer review and any attached files.

Reviewer #1: No

Reviewer #2: Yes: Penglin Sun

Reviewer #3: No

---

## [Author Response · Author response to Decision Letter 0]

18 Feb 2020

Dear Dr Sean Michael Prager,

Please find enclosed our revised manuscript entitled “A diagnostic real-time PCR assay for the rapid identification of the tomato-potato psyllid, Bactericera cockerelli (Šulc, 1909) and development of a psyllid barcoding database”. We thank you and the reviewers for your careful review of our submitted manuscript and the opportunity to resubmit an improved version. We find the comments to be fair and constructive and have helped to improve the final manuscript. Please see below our responses to points raised by the reviewer’s comments and the amendments we have made to the final manuscript. We provide a copy of the resubmitted manuscript with track changes and track changes accepted. Line numbers refer to those given in the resubmitted manuscript with track changes accepted.

We hope that you will consider this revised manuscript of a high enough standard to be published in PLOS ONE. 

Yours Sincerely,

Dr Jason C Sumner-Kalkun

(on behalf of all co-authors)

• Thank you for considering our work for publication in your journal. We found all reviewer comments to be useful and constructive and appreciate you overall assessment of the manuscript. We are pleased to have the opportunity to resubmit an improved version. We have made considerable efforts to condense the introduction and discussion section to include only relevant information and to streamline the manuscript. We agree that there was some duplication and repetition in the discussion, and it has been re-written accordingly. We decided, due to the technical nature of the paper that a separate results and discussion section was more appropriate. We have addressed the reviewer concerns regarding data availability and all sequence data has been uploaded to GenBank and is now free to be made publicly available. We hope that our amendments are deemed adequate to meet the high standards of PLOS ONE and are excited about the possibility of publishing with you. 

REVIEWER COMMENTS TO AUTHOR

Reviewer #1: This manuscript describes a qPCR assay to identify potato psyllid intercepted in shipments. The assay is paramount to Europe's ability to detect potential introductions of this psyllid, which would be harmful to agricultural production. The authors describe the assay and confirmed that it does not amplify the ITS gene of other psyllids. My major concern for the manuscript is that it is overwritten and over-interpreted. The study is very simple - qPCR assay to detect potato psyllid - yet the text is over 50 pages long, includes unrelated information in the introduction, and includes an overly long discussion. The manuscript should be re-written to focus only on the assay and its use in trade commodities. Specific comments are provided in an attached document. I will apologize for my handwriting.

MAJOR POINTS

We appreciate your thorough assessment of our manuscript and thank you for your time. We found your comments very constructive and helpful. We have taken the care to reduce the introduction and discussion sections considerably to provide more focus on the assay and its uses, removing a lot of the duplication. The manuscript has been edited down to 31 pages + supplementary material. We attempted to produce a combined results and discussion section but felt that, due to the technical nature of the paper, keeping these separate was preferable. We hope that you will agree with this assessment on reading the improved version.

On the recommendation of the reviewer on line 486 of the previous manuscript we have performed the assay on Potato DNA to check for cross-reaction. No false positives were obtained from 8x reps of 3 Potato samples “Maris Piper” variety. 

MINOR POINTS

1. Line 45: Abstract overwritten, stats to be removed, word count reduced

- The abstract Line 21-39 has been reduced in size with all stats removed and is now within the word limit (252 words)

2. Line 47: Remove “-“ in “Potato-Psyllids”

- Changed to “Potato Psyllid” now line 41 

3. Line 49: “The feeding of….” To be changed to “Feeding by”

- Changed as suggested now line 43

4. Line 53: Psyllid yellows refers to the feeding damage described above.

- Removed to avoid confusion and improve accuracy. Line 47

5. Lines 55-56: Change “…is also able to reproduce on…” to “…can also complete development on species of….”

- Changed as suggested line 49-50

6. Lines 56-58: Statement not deemed true

- Statement removed line 51

7. Line 61: Remove statement on Lso transmission to non-host plants of B. cockerelli

- Statement removed line 53

8. Line 64-65: Remove claims about B.cockerelli populations observed to differ in their ability to spread Lso

- Changed to: “Evidence suggests that these genetic types may differ in their ability to spread Lso…” Lines 56-57

9. Line 86: Haplotype B is also found in Bactericera maculipennis

- Information added to the text line 71

10. Line 111: typo capsicum not italics

- Changed to “…Capsicum…” line 84

11. Line 223:This table is referenced a lot, make it a real table

- Supp Tab. S1 now changed to Table 1. In results section Line 244-250. Cited on lines: 244. Supp Tabs 2-4 renumbered to Supp Tabs 1-3 and Tables 1-3 renumbered to Tables 2-4. 

12. Line 276: Submitted to NCBI? Provide accession numbers

- Accession numbers added to Table 1. Lines 246-252 and Table 4. Lines 271-276

13. Line 314-315: change “….cloned DNA as mentioned below.” To “..DNA below”.

- Changed to “….cloned DNA (see below).” Line 290

14. Line 319: change “immatures” to “nymphs”

- The term “immatures” is preferred by leading psyllid taxonomists Daniel Burckhardt and David Ouvrard, that latter of whom is an author on this paper. See ref: (Burckhardt et al. 2014). We have kept the term “immatures” or “immature life stages” throughout.

Burckhardt D, Ouvrard D, Queiroz D, Percy D (2014) Psyllid Host-Plants (Hemiptera: Psylloidea): Resolving a Semantic Problem. Florida Entomol 97:242–246 . https://doi.org/10.1653/024.097.0132

15. Line 411: “…Bactericera…” to be italicised

- Changed to italics. Line 354

16. Lines 439-441: Section to be re-written as inaccurate wording used

- This section was removed in the re-write of the discussion. 

17. Line 468: Suggestion to perform further validation on Solanaceous DNA

- 3 x samples of Solanum tuberosum ‘Maris Piper’ were tested and were negative results added to lines: 194-196 and 284-285. Also results of primer blast etc. did not return any hits for Solanum species or any plant sequences. 

Reviewer #2

- We are thankful to the reviewer for their detailed and careful examination of our paper. They have provided very useful, constructive comments regarding the technical aspects of the paper and have informed us of errors in the finer details. We hope we have incorporated changes to their satisfaction, and we have endeavoured to clear up the technical details that were missing or incorrect. 

1. Data availability

- Psyllid DNA sequences have been uploaded to GenBank and accession numbers are provided in Tab1. And Tab4; lines 246-252 and 271-276 respectively. 

2. Page 8 Line 163: What part of the body is used for micro-dissection to extract DNA? The authors should describe the micro-dissection procedure in more detail rather than only citing the papers.

- The non-destructive DNA extraction method is described on lines 121 – 132. “Micro-dissection” was used here to describe the piercing of the abdomen and thorax. “Micro-dissection” has been changed to “pierced” as a more appropriate term (line 126).

3. Page 8 Line 172: “For amplification of ITS2 primers CA55p8sFcm-F and CA28sB1d-R [60] and for amplification of CO1 gene regions arthropod barcoding Primers LCO1490 and HCO2198 [61].” The authors should check the grammar here. It is not a complete sentence. It could be “For amplification of ITS2, primers CA55p8sFcm-F and CA28sB1d-R [60] were used, and for amplification of CO1 gene regions, arthropod barcoding Primers LCO1490 and HCO2198 [61] were used.”

- Changed as suggested lines 135- 137

4. Page 10 Line 204: “DNA was extracted as above using the non-destructive method, amplified and cloned into competent Escherichia coli cells using the TOPO TA cloning kit (Thermo-Fisher).” The authors should specify what genes (ITS2 or CO1?) they amplified for cloning, and what restriction enzyme (EcoRI?) they used to linearize the plasmid.

- Information added and moved from later section 2.5.2 Sensitivity. Now line 171-178

5. Page 10 Line 212: The authors need to list the real time PCR cycling conditions here, for example XX degrees for XX seconds.

- Added lines 178-181

6. Page 10 Line 223: “All reactions with non-target psyllid DNA were run in conjunction with a TaqMan Exogenous Internal Positive Control Reagent Kit (Applied Biosystems) to ensure false positives were not obtained due to inhibition within the reaction”. Here, “ensure” should be “rule out the possibility that”

- Changed as suggested lines 196-201

7. Page 11 Line 226: “DNA from all non-target psyllids was sequenced to ensure psyllid DNA was present in all reactions to rule out false negatives due to inefficient DNA extraction.” What DNA was sequenced? PCR product from ITS2 or CO1? The authors need to specify

- Details now added to new Tab 1 and citation to table included on lines 246-252

8. Page 11 Line 239: “6 subsequent dilutions were made. Stock DNA 10 ng/μl was linearised using EcoRI restrictions enzyme (New England Biolabs),” Here “6 subsequent dilutions” should be “8 subsequent dilutions”, according to the nine point 10-fold dilution series mentioned on Page 11 Line 236.

- Corrected Line 212-213

9. Page 12 Line 252: “A six point 1:10 dilution series starting at 10ng/μl was used with each dilution being performed in triplicate.” Here, “six point” should be “nine point” according to Page 11 Line 236.

- Only 6 points were used for repeatability. This is sufficient to analyse standard curves between runs. Lines 222-223 refer to sensitivity experiments only.

10. Page 12 Line 263: “For each tested parameter, optimization was performed across an eight point 1:10 dilution series starting at 10ng DNA.” Here, “eight point” should be “nine point”, “10ng” should be “10ng/μl”.

- Corrected. Line 234

11. In Supplementary table S1, green and red color coding should be explained in the text. What does TBC mean? Accession numbers should be given for all the sequences. Accession numbers in Table 3 should also be given and TBC should be explained.

- We apologise for this error; this colouring has been removed as was an artefact of preparing the table and shouldn’t have been included in the submitted version. TBC was used to show we were waiting for accession numbers. Accession numbers are now added to tables and TBC removed. Tab. 1 lines: 246-247 Tab.4 lines:

12. Page 14 Line 289: “CO1 genes showed higher similarity and generally less conserved and variable regions compared to ITS2 regions.” Here “less conserved and variable” should be “less variable”.

- Corrected line 266

13. Page 17 Line 310: “0.2 µ/mol” should be “0.2 µM”.

- Corrected line 285

14. Page 18 Line 324: “The copy number calculator available at http://scienceprimer.com/copy-number-calculator-for-realtime-pcr was used.” Here a hyperlink should be created. According to the link and the formula given, 0.00001ng DNA equals 4.879×10000 copies, if length of gene region is considered 187bp (product length of ITS2 in real time PCR). However, the authors calculated that it equals to 200bp. Please double check the calculation.

- Limit of detection is actually 0.000001 ng DNA. This mistake of 10 fold higher amounts stated in the text was found throughout and in tables. We have now corrected them. The correct equation should be:

Number of Copies = (ng DNA(0.000001) x 6.022x1023) ÷ ((length of plasmid 4656bp + cloned fragment 700bp) * 1x109 * 660) = 170.36 copy numbers. 

15. Page 18 Line 337: “At primer concentration, 0.5 μM the assay was less sensitive only amplifying up to 0.001 ng DNA.” It should be “At primer concentration 0.5 μM, the assay was less sensitive only amplifying up to 0.001 ng DNA.”

- Corrected. Lines 313-314

16. Page 18 Line 338: “At higher primer concentrations (0.5 and 1.0) the assay showed higher sensitivity” Here “(0.5 and 1.0)” should be “(1.0 μM)”.

- Corrected. Line 314

17. Page 19 Line 350: “The precision of the assay was lower at higher MgCl2 concentrations 6mM and 8mM (Supp Tab. S3).” Here “6mM and 8mM” should be “7.5mM and 9mM”.

- Corrected. Lines 326-327

18. Page 19 Line 354: “Reactions at 58 °C were 10 to 100-fold less sensitive than reactions at 58 °C.” Here it should be “Reactions at 58 °C were 10 to 100-fold less sensitive than reactions at 64 °C.”

- Corrected. Lines 330-331

19. Page 20 Line 367: “We have designed and validated the first species-specific, qualitative real-time PCR TaqMan assay for B. cockerelli by using the comparison of 73 non-target species to identify unique gene regions that were suitable for primer/probe design and species differentiation.” Here “qualitative” should be “quantitative”.

- Changed to quantitative. Line 351

Reviewer #3

- We thank the reviewer for their thoughtful assessment of our manuscript and are pleased that only minor corrections were found throughout. The corrections have improved the manuscript greatly and have ironed out some important technical errors. We hope that our amendments are deemed satisfactory and have covered the issues they have raised.

1. Page 8 line 175: please replace amount of primers used with final concentration of primers (or add this)

- Added. Line 138

2. Page 10 line 213: please add cycling conditions of real time PCR, as done for CO1 and ITS2 amplification

- Added lines 178-181

3. Table 1: should include also B. cockerelli, so to include fragment size of amplicons for this species. In alternative, fragment sizes can be added to the main text

- B. cockerelli added to table 2. Line 262-263

4. Table 3: not clear what the "/" symbol in the CO1 column means

- Samples with / were not amplified in this region. Accession numbers for each sample have been added and this is explained better in the text. Lines: 252 Tab.1 ; 276 Tab. 4

5. Page 17 line 310: please check spelling of concentration

- Corrected to µM. Line 285

6. Page 17 line 316: numbers seem not to add up: how many technical replicates were used per sample?

- Information on technical reps is incorporated into table 4. Some samples were tested in duplicate, triplicate or 6x replicates.

7. Page 18 line 323: I have tried the formula myself using the concentration (0.00001 ng) and fragment size (187 bp) specified by the authors, but I get a quite different number of ITS2 copies (about 50,000 versus 200). Please double check, and add actual numbers to the formula.

- Limit of detection is actually 0.000001 ng DNA. This mistake of 10-fold higher amounts stated in the text was found throughout and in tables. We have now corrected them. The correct equation should be:

Number of Copies = (ng DNA(0.000001) x 6.022x1023) ÷ ((length of plasmid 4656bp + cloned fragment 700bp) * 1x109 * 660) = 170.36 copy numbers. 

8. Of some concern is the author's answer to the data accessibility question. Authors stated that they are not going to make all data available, with a generic "Some restrictions will apply". Please explain what data will not be made accessible and why.

- All data will be made available. Accession numbers were not available at the time of submission as they were restricted by one or more of our projects until we had consent to upload them to public databases.

---

## [Decision Letter · Decision Letter 1]

3 Mar 2020

PONE-D-19-35047R1

A diagnostic real-time PCR assay for the rapid identification of the tomato-potato psyllid, Bactericera cockerelli (Šulc, 1909) and development of a psyllid barcoding database.

PLOS ONE

Dear Dr. Sumner-Kalkun,

Thank you for submitting your manuscript to PLOS ONE. After careful consideration, we feel that it has merit but does not fully meet PLOS ONE’s publication criteria as it currently stands. Therefore, we invite you to submit a revised version of the manuscript that addresses the points raised during the review process.

I appreciate the effort you have taken in both addressing the reviewer concerns and revising this work. In that context, I have rendered a decision of minor changes. This is primarily to give you the opportunity to make or address the additional reviewer comments.

We would appreciate receiving your revised manuscript by Apr 17 2020 11:59PM. To enhance the reproducibility of your results, we recommend that if applicable you deposit your laboratory protocols in protocols.io, where a protocol can be assigned its own identifier (DOI) such that it can be cited independently in the future. For instructions see: http://journals.plos.org/plosone/s/submission-guidelines#loc-laboratory-protocols

We look forward to receiving your revised manuscript.

Kind regards,

Sean Michael Prager, Ph.D.

Academic Editor

PLOS ONE

Reviewers' comments:

Reviewer's Responses to Questions

**Comments to the Author**

1. If the authors have adequately addressed your comments raised in a previous round of review and you feel that this manuscript is now acceptable for publication, you may indicate that here to bypass the “Comments to the Author” section, enter your conflict of interest statement in the “Confidential to Editor” section, and submit your "Accept" recommendation.

Reviewer #1: (No Response)

2. Is the manuscript technically sound, and do the data support the conclusions?

Reviewer #1: Yes

3. Has the statistical analysis been performed appropriately and rigorously? 

Reviewer #1: Yes

4. Have the authors made all data underlying the findings in their manuscript fully available?

Reviewer #1: Yes

5. Is the manuscript presented in an intelligible fashion and written in standard English?

Reviewer #1: Yes

6. Review Comments to the Author

Reviewer #1: The revised version of this manuscript is a substantial improvement from the original submission. I have only a handfull of minor suggestions in the attached PDF that the authors might consider.

7. PLOS authors have the option to publish the peer review history of their article (what does this mean?). If published, this will include your full peer review and any attached files.

Reviewer #1: No

---

## [Author Response · Author response to Decision Letter 1]

6 Mar 2020

REVIEWER COMMENTS TO AUTHOR

Reviewer #1: 

The revised version of this manuscript is a substantial improvement from the original submission. I have only a handful of minor suggestions in the attached PDF that the authors might consider.

Response:

We thank the reviewer for their careful assessment of the manuscript and the comments previously made to help us to improve and streamline the manuscript. We have incorporated most minor changes and edits as detailed below.

MINOR POINTS

Line 21: Remove “many”

- Removed. Line 21 (of document with track changes accepted).

Line 24: You can keep central america if you want to, but technically, Central America is a region within North America, so all you really need to say is "North America"

- Reduced to just “North America”. Line 24

Line 25: add “and is considered an threat for introduction in Europe and other pest-free regions.”

- Added: “…; and is considered a threat for introduction in Europe and other pest-free regions.” Lines 24- 25

Lines 27-30: remove “successfully” and restructure sentence to remove “(100% n=X)” after each sample type.

- Sentence restructured and percentages and sample numbers removed. Lines 27-30.

Lines 58-61: I think you need to find a way to combine this short paragraph with the preceding paragraph.

- Paragraph added to a previous paragraph. Lines 48-51

Lines 67-72: Since the manuscript focusses on potato psyllid, I don't think you need such an in-depth discussion about the various Liberibacter haplotypes

- We would prefer to keep the information about Lso haplotypes in as we believe it is an important aspect of Lso epidemiology. The information on Lso haplotypes is greatly reduced compared to the previous manuscripts.

Line 77: change “North-West” to “Northwest”

- Changed. Line 76

Line 108: Change “North-Western” and “South-Western” to “Northwestern” and “Southwestern”.

- Changed. Line 107

Line 114: US collections of non-targets isn't mentioned in abstract.

- Specimens were collected by Andy Jensen from multiple locations in the USA and tested with the assays as detailed in Table 1. We have added “USA” to the abstract. Line 32.

Lines 129-132: “DNA extraction in DNeasy Blood and Tissue Kit Protocol from Animal Tissues (Qiagen)” mentioned previously on line 128? 

- Information condensed to avoid duplication. Lines 128-130

---

## [Editor Report · Decision Letter 2]

9 Mar 2020

A diagnostic real-time PCR assay for the rapid identification of the tomato-potato psyllid, Bactericera cockerelli (Šulc, 1909) and development of a psyllid barcoding database.

PONE-D-19-35047R2

Dear Dr. Sumner-Kalkun,

We are pleased to inform you that your manuscript has been judged scientifically suitable for publication and will be formally accepted for publication once it complies with all outstanding technical requirements.

With kind regards,

Sean Michael Prager, Ph.D.

Academic Editor

PLOS ONE
---

## [Editor Report · Acceptance letter]

10 Mar 2020

PONE-D-19-35047R2 

A diagnostic real-time PCR assay for the rapid identification of the tomato-potato psyllid, *Bactericera cockerelli* (Šulc, 1909) and development of a psyllid barcoding database. 

Dear Dr. Sumner-Kalkun:

I am pleased to inform you that your manuscript has been deemed suitable for publication in PLOS ONE. Congratulations! Your manuscript is now with our production department. 

With kind regards,

on behalf of

Dr. Sean Michael Prager 

Academic Editor

PLOS ONE